# Newborn Care Practices and Associated Factors Influencing Their Health in a Northern Rural India

**DOI:** 10.3390/children10020408

**Published:** 2023-02-20

**Authors:** Md Arfin Islam, Md Suhail Khan, Anas Ahmad Khan, Bayapa Reddy Narapureddy, Kalyan Viswanath Reddy Lingala, Nazim Nasir, Khursheed Muzammil, Irfan Ahmad, Adam Dawria, Ahmed Faheem, Ali Mohieldin

**Affiliations:** 1Department of Community Medicine, IIMS & R, Lucknow 226026, India; 2Department of Public Health, CAMS, Khamis Mushait Campus, King Khalid University, Abha 62529, Saudi Arabia; 3Department of Community Medicine, United Institute of Medical Sciences, Prayagraj 231313, India; 4Department of Public Health, College of Health Sciences, Saudi Electronic University, Abha 62529, Saudi Arabia; 5Department of Basic Medical Sciences, CAMS, Khamis Mushait Campus, King Khalid University, Abha 62529, Saudi Arabia; 6Department of Clinical Laboratory Sciences, CAMS Abha, King Khalid University, Abha 62529, Saudi Arabia

**Keywords:** new-born, cord care, new-born bathing, home deliveries, institutional deliveries, birth attendants, breastfeeding

## Abstract

Introduction: In developing countries, neonatal mortality is the most neglected health issue by the health system, leading to its emergence as a public health problem. A study was undertaken to assess the influence of factors and newborn care practices influencing newborn health in the rural area of Bareilly district. Methodology: The descriptive cross-sectional study was organized in the rural areas of Bareilly. Study participants were selected based on the mothers who gave birth to a baby during the last six months. The mothers who delivered in that area within six months were included and, using the semi-structured questionnaire, data were collected. Data were analyzed using Microsoft Excel and SPSS 2021 version for windows. Results: Out of 300 deliveries, nearly one-quarter of the deliveries, 66 (22%), were happening in homes, and most of the deliveries, 234 (78%), happened in hospitals. It was observed that unsafe cord care practices were observed more among nuclear families, 8 (53.4%), than joint families, 7 (46.6%), and it was found to be statistically insignificant. The Unsafe feed was given 48 (72.7%) more commonly among home deliveries than institutional deliveries 56 (23.9%). Mothers’ initiation of delayed breastfeeding was nearly the same in both home and hospital deliveries. Delayed bathing was observed in nearly three-fourths of mothers, 125 (70.1%), aged 24–29 years, followed by 29 (16.8%) in the age period of 30–35 years. Conclusion: The practice of essential newborn care still needs to improve in Bareilly; there is a need to create awareness among the mothers and family members on newborn and early neonatal care aspects, such as promoting exclusive and early initiation of breastfeeding and delayed bathing practices.

## 1. Introduction

Universally, in the last two decades, neonatal mortality has dropped almost 51 percent; the neonatal mortality rate in 1990 was 37/1000 live births, which dropped to approximately 18/1000 live births in 2021 [1]. In 2021, 2.3 million neonates have died in the first four weeks of life, roughly 6400 neonatal deaths per day. However, the reduction in neonatal mortality has decelerated from 1990 to 2021 compared to infant mortality and under-five mortality during the same period [1]. The neonatal mortality rates remain static, especially in non-industrialized countries [2].

India contributes 17.7% of the world’s population and nearly one-sixth of the total live births [3]. Neonatal deaths in India have declined by nearly three-fourths in the last five decades from 83.6/1000 live births in 1971 to 20.3/1000 live births in 2020 [4]. The deceleration of the neonatal mortality rate that accelerated in the recent decade is due to the introduction of the neonatal resuscitation program, and the National Health Mission program to address maternal and child care, including neonatal care [5]. Some studies mention the care during the early neonatal period and newborn care practices on day one after delivery to the lower neonatal risk of morbidity and mortality [6].

Newborn care practice interventions can prevent neonatal deaths. Quality care during antenatal and perinatal periods are essential for reducing neonatal mortality [6,7]. World Health Organization (WHO) recommended preventing neonatal mortality through the preparation of mothers for managing complications, adequate good quality antenatal care, practicing clean delivery, and cord care, to avoid hypothermia through the skilled care of mothers such as thermal care, early and exclusive breastfeeding, and immunizing the child [8,9]. Some societies lack the familiarity and exercise of newborn care, for instance, the feeding of colostrum, prevention of hypothermia, and exclusive breastfeeding; even awareness about care seeking on the identification of life-threatening signs was very poor [10].

Though the antenatal tetanus toxoid immunization was implemented long before as a cost-effective solution to prevent neonatal tetanus, skilled attendance during delivery, clean cord care, early initiation of colostrum, and exclusive breastfeeding, the neonatal mortality rate has changed relatively little [11]. The WHO recommends improving essential newborn nurturing practices at birth to lessen neonatal morbidity and mortality [12]. In developing countries, neonatal mortality is the most neglected health issue by the health system leading to its emergence as a public health problem. In India, adverse consequences on the child survival rate were due to different cultural beliefs and practices among the age-old traditional populations [13]. The essential basic and emergency obstetric and newborn care interventions were introduced to reduce under-five mortality, but do not yet fully address the roots [14]. With this background, this study was undertaken to determine the attributing factors and newborn care practices influencing newborn health in the rural areas of Bareilly district.

## 2. Materials & Methods

Study Settings: The current study was planned to be conducted in the rural area of Bareilly, where the Shri Ram Murti Smarak Institute of Medical Sciences (SRMSIMS) delivers services.

Study Design: An analytical cross-sectional study was conducted from 1 August 2021 to 31 July 2022 on all the mothers who had a live baby aged below six months and residing in the selected rural areas of the Bareilly district of Uttar Pradesh.

Sampling: Considering the rate of commencement of early breastfeeding, 43% within one hour after birth in India (NFHS-4 2015-16) [15], the sample size was determined using allowable error of 15% of prevalence after the substitution of values, n = 4pq/d^2^, *p* = 43 (initiation rate of colostrum within the first hour of life), q= (100-*p*) = 57, d =relative permissible error = 15% of *p*, n = sample size, (n = 239). Considering a 10% rate of non-response (24), the sample size came out to be 263 and finally we studied 300. Three hundred mothers that had a live baby of an age below six months in selected villages made the estimated sample size.

## 3. Methods

The data was collected from the participants by making personal visits to the household having eligible mothers; after explaining the purpose and objectives of the study to all post-natal mothers, voluntary informed consent was obtained. The data was obtained by filling out the validated pre-designed questionnaire through face-to-face interviews with eligible mothers who had a live baby of an age below six months and residing in the selected villages. The study tool included a socio-demographic profile, the utilization of maternal health services, post-natal newborn care knowledge, and practices.


**Inclusion Criteria**


All the mothers who gave birth within the last six months to a live babyMothers willing to participate in the study.


**Exclusion Criteria**


Mothers who delivered and stayed outside of the study area.Mothers who were seriously ill and hospitalized.


**Statistical Analysis**


The Data was entered into Microsoft Excel 2021 software version for windows and imported to SPSS-21 software, and statistical analysis was performed (Statistical Package for Social Science software version-21 IBM Corporation, Armonk, NY, USA). Categorical variables were expressed in frequencies and proportions. A Chi-square and Fisher’s exact test established the association between qualitative variables.


**Ethical Consideration**


The study obtained institutional ethical committee approval (IEC/SRMS/2021/120/3 dated 18 November 2021) from the ethical committee of SRMS Institute of Medical Sciences, Bareilly.

## 4. Results

Nearly three-fourths, 212 (70.8%), of post-natal mothers were in the 24–29 years age group, followed by 58 (19.3%) in the 30–34 age group. Approximately 25.6 years was the mean age of the mother. Nearly half of the mothers were illiterate, 134 (44.7%), while almost one-fourth of all mothers, 76 (24.07%), had a high school education and above. About two-thirds of mothers belonged to the lower middle class, 208 (66.0%), followed by 70 (22.0%) belonging to the lower class, as per the modified BG Prasad classification. The majority of mothers, 129 (37.3%), had delivered their second child, while 8.7% had delivered four and above birth order. Details have been provided in Table 1.

Regarding the associated factors influencing newborn health, this study observed that almost 95% of participants practiced safe cord care. An approximately equal number of participants initiated early (within an hour after delivery), 149 (49.6%), and delayed, 151 (50.4%), breastfeeding. Most of the participants, 177 (59%), were practicing delayed baths for newborns. (Figure 1)

Out of 300 deliveries, nearly one-quarter of the deliveries, 66 (22%), were happening in homes, and most of the deliveries, 234 (78%), happened in hospitals. Out of 66 home deliveries, in the majority of deliveries, 51 (77.4%) of the birth attendants used the new blade to cut the umbilical cord. In some deliveries with the non-availability of new blades, the birth attendants used the old blades, 10 (15.1%), and some used a knife, 3 (4.5%). In contrast, those deliveries happened in all institutions; most used either sterilized scissors, 197 (84.1%), or new blades, 37 (15.9%). More than one-quarter of the babies, 18 (27.3%), were given baths immediately after delivery. Whereas in institutional deliveries, in one-third of the babies, 82 (35%), the first bath was done within 24–48 h, and two-thirds of babies were given the first bath after 48 h. Pre-lacteal feeds such as Ghutti (15.1%) and honey (16.7%) provided more in-home deliveries compared to institutional deliveries (2.39% and 0.83%, respectively). The difference between the place of delivery with new-born practices was observed to be statistically significant (Table 2).

Safe cord care: in the case of institutional delivery, most mothers reported cutting the cord with sterilized scissors or the blade used, 206 (72.4%) belonging to the age group 24–29 years, followed by 54 (18.9%) aged 30–35 years; the association of cord cutting practices with the age of the mother was found to be significant. Safe cord-cutting practices were found to be more predominant among 213 (74.7%) semi-skilled laborers than 45 (21.1%) skilled laborers, and it was highly significant (*p* = 0.000) statistically. Safe cord-cutting practices were found to be significantly higher, 125 (43.8%), among women having parity two, followed by mothers having single parity, 86 (30.1%) (Table 3).

It is evident from the Table 4 that among mothers practicing early breastfeeding, it was more commonly observed in the age group of 24–29 years, 78 (52.3%), followed by 30–35 years, 48 (32.3%), and it was found to be significant statistically (*p* < 0.05). Delayed breastfeeding was observed more among mothers belonging to joint families, 92 (61.0%), than nuclear families, 59 (39.0%), which was significant statistically. Delayed breastfeeding was more commonly observed among semi-skilled mothers, 115 (76.1%), followed by mothers engaged in skilled occupations, 22 (14.5%). Among mothers practicing delayed breastfeeding, it was more commonly observed among parity two, 60 (39.8%), compared to mothers having single equality, 58 (38.4%), and it was significant.

It can be observed from Table 5 that the practice of delayed bathing was noticed in 125 (70.1%) mothers in the 24–29 years age group, followed by 29 (16.8%) in the 30–35 years age group. Among 177 mothers practicing delayed new-born bathing practices, 112 (62.6%) of them belonged to joint families as compared to 65 (36.4%) nuclear families, and this difference in bathing practices among the nuclear and joint families was found to be significant (*p* = 0.03). Among the 123 mothers practicing the immediate bathing of a new-born, most of them were illiterate mothers, 87 (70.7%), compared to mothers having a primary education, 12 (9.7%), and it was found the the difference was statistically significant. A greater number of mothers, 73 (41.2%), having parity two were found to be practicing delayed bathing, followed by mothers having single parity, 64 (36.1%), and it was found that the difference was statistically significant.

## 5. Discussion

In the present study, most of the study participants were between 24 to 29 years of age, the majority were staying as joint families, and most were illiterates. A study conducted in Ethiopia by Semanew Y et al. [16] also observed that the majority of the participants were between the age of 21 to 35 years. Regarding educational status, most of the participants were either illiterate or just able to read and write. Regarding the occupation, most of the participants were housewives. The Ethiopia study also monitored similar study groups. The intranatal care, such as using a new blade/sterile to cut the cord, is observed in more than three-fourths, 51 (77.2%), of home deliveries. The results aligned with NFHS-4 data [15]. A disposable delivery kit (DDK) with the clean blade to cut the cord care was observed in only 33% of cases and the Rahi et al. [14] studies conducted in Delhi observed that a new blade was used in 78.3% of cases to cut the cord of home deliveries [16]. In contrast, a study by Otolorin et al. [17] and Sartaj Ahmad et al. [18] in Meerut showed that a new blade was used in a comparatively lower percentage (63.82%) of home deliveries. Another study conducted in Ethiopia by Semanew et al. [16] observed that nearly half of the deliveries used the new or boiled blade [16]. In this study, old blades and unsterile knives were used in 5 (7.6%) and 3 (4.5%) cases, which is higher compared to the results of the Manju Rahi et al. [14] study where they observed that old blades and knives were used in only 2.2% and 2.1% of home deliveries, respectively. On the contrary, the overall use of old blades and knives, 21.28%, was higher in a study by Sartaj Ahmad et al. [18].

It was found that most of the newborns, 27.3%, were bathed within one hour after birth with warm water and dried with a clean cloth and 21.3% were wiped within 24 h after delivery; nearly one third, 37.8%, were given a delayed bath in home deliveries, which is in alignment with the Manju Rahi et al. [14] studies which noted that a large percentage, 82.6%, of home delivered babies were bathed immediately after birth and 59% had delayed bathing. Similar results were noted by Vijayalakshmi S et al. [19] conducted in rural Puducherry, India, where 70.6% of newborn babies get bathed after 24 h of life [19]; another study conducted by Bekele K et al. observed that 35% of the babies received bathing within 24 h and the remaining 65% received bathing after 24 h [20]. Bathing babies within the first hour of life was more common with the 24–35 years age group, illiterates, and joint families. The early bathing of newborns significantly increases hypothermia among those babies, which was demonstrated in a randomized controlled trial conducted in Uganda [21], so there is a need to encourage the delayed bathing of the babies.

In the current study, it was observed that breastfeeding began within 4 h in 32 (42.4%) of home deliveries compared to 117 (49.9%) institutional deliveries. The delayed initiation of breastfeeding is most observed in illiterates, backward class communities, lower-middle-class society, and housewives. The study’s findings are comparable to Sonia Puri et al. [22]; studies in the urban slums of Chandigarh revealed that, out of 270 respondents, 58.9% initiated breastfeeding within 6 h of birth, 40.0% mothers gave pre-lacteal feed, and 15.9% had discarded the colostrum. Semanew Y et al. [16] observed that nearly 84% of the babies received breastfeeding within an hour of birth. In the home deliveries, only one-quarter of the babies, 18 (27.3%), received breast milk as the first feed after birth, which is much less when compared with 178 (76.0%) and institutional deliveries. Semanew Y et al. [16] and Bekele K et al. [20] studies observed that about 97% and 73% of the babies received breast milk as the first feed, respectively. The study results are comparable to Manju Rahi et al. [14], where it was observed that breast milk was given as a first feed in 27 (32.9%) newborns. Pre-lacteal feeds, such as honey/sugar water (16.7%) and ghutti (15.1%), were given (2.1% and 0.8%, respectively) more in in-home deliveries compared to hospital deliveries. The practice of feeding pre-lacteal foods is prevalent in India, as reported in studies conducted by Singh MB et al. [23] and Deshpande et al. [24].

The overall knowledge about neonatal danger signs and health care seeking was poor among mothers. A similar study conducted in China by Zhou J et al. [25] also measured the good knowledge of danger signs and associated factors among newborns was only 50%.

## 6. Conclusions

There was a significant difference in essential newborn care and factors influencing the hospital and home-delivered mothers. Most of them used the new blade/sterile instrument to cut the cord in hospital deliveries, more so than in home deliveries. Most newborns were given delayed bathing after birth in institutional deliveries, whereas in-home delivery practices, such as a delay in the initiation of breastfeeding and providing pre-lacteal feeds, were very commonly observed. This implies that there are still many lacunae in the clean delivery practices and the community’s early neonatal care. The Govt. of India initiated several programs to improve maternal and child health, to decrease the neonatal deaths; still, it has not reached the needy people. The creation of awareness about the programs such as National Health Mission, Janani surakha yojan to promote institutional deliveries, Janani Sishu Suraksha Karyakram, Newborn action plan to improve the neonatal care, etc., and the benefits of the programs need to be propagated within the public to utilize the available services.

## Figures and Tables

**Figure 1 children-10-00408-f001:**
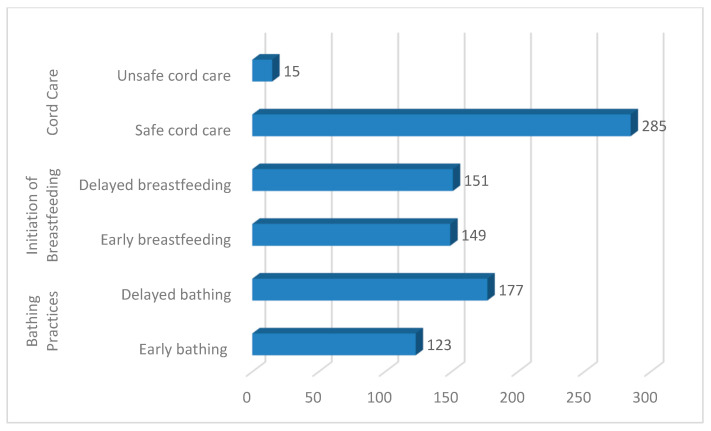
Frequency distribution of factors influencing the newborn care practices.

**Table 1 children-10-00408-t001:** Distribution of study participants based on demographic details.

Socio-Demographic Variables	Number	Percentage (%)
**Age (years)**		
18–23	23	7.6
24–29	212	70.8
30–35	58	19.3
>35	7	2.3
Total	300	100.0
**Religion**		
Hindu	87	29.0
Muslim	213	71.0
**Family Type**		
Joint family	204	68.0
Nuclear family	96	32.0
Total	300	100.0
**Education of mothers**		
Illiterate	134	44.7
Primary	38	13.6
Junior High	51	17.0
High school	63	21.0
Intermediate	11	3.6
Graduate or above	3	01
Total	300	100.0
**Occupation**		
Unskilled labor	21	7.0
Semi-skilled labor	217	72.3
Skilled	48	16.0
Shopkeeper/Clerical	14	4.7
Total	300	100.0
**Socio-economic class**		
Upper class	5	1.6
Upper middle class	6	2.0
Middle class	26	8.6
Lower middle class	193	64.4
Lower class	70	23.4
Total	300	100.0
**Parity**		
1	89	29.6
2	129	43.2
3	56	18.6
>4	26	8.6
Total	300	100.0

**Table 2 children-10-00408-t002:** Distribution of new-born according to practices concerning the place of delivery.

New-Born According to Practices		Number of Home Deliveries n = 66	Number of Institutional Deliveries n = 234	Chi-Square Value, df	*p*-Value
**Cord Care**					
**Safe cord care (n = 285)**	New Blade/sterilized scissors	51 (77.4)	234 (100)	-	-
**Unsafe cord care (n = 15)**	Old Blade	10 (15.1)	0 (0.0)		
	Knife	3 (4.5)	0 (0.0)		
	Scissor	2 (3.0)	0 (0.0)		
**Bathing practice**					
**Early bathing (n = 123)**	within 1 h	18 (27.3)	0 (0.0)	15.60, -	-
	2–24 h	14 (21.3)	0 (0.0)		
	24–48 h	9 (13.6)	82 (35.0)		
**Delayed bathing (n = 177)**	>48 h	25 (37.8)	152 (65.0)		
**The first feed of baby**					
**Safe first feed (n = 196)**	Breast milk	18 (27.3)	178 (76.0)	54.11, df-1	0.00
**Unsafe first feed (n = 104)**	Water	11 (16.7)	15 (6.4)		
	Honey	11 (16.7)	5 (2.3)		
	Ghutti	10 (15.1)	2 (0.8)		
	Cow milk	16 (24.2)	34 (14.5)		
**Timing of commencement of breastfeeding**					
**Early breastfeeding** **(n = 149)**	Less than an hour after birth	10 (15.1)	46 (19.8)	0.04, df-1	0.82
	Within 1–4 h	22 (27.5)	71 (30.0)		
**Delayed breastfeeding** **(n = 151)**	Within a day	17 (25.7)	57 (24.3)		
	After a day	12 (18.2)	40 (17.0)		
	After three days	5 (7.5)	20 (8.5)		

**Table 3 children-10-00408-t003:** Socio-demographic factors influencing Cord care practices.

Socio-Demographic Variables	Cord Care					
	Safe Cord Care n = 285		Unsafe Cord Care n = 15		*p*-Value	Chi-Square Value, Degree of Freedom
**Age of mother** **(in years)**	**No.**	**(%)**	**No.**	**(%)**	0.003	13.86, df-3
18–23	20	7.2	3	20.0		
24–29	206	72.1	6	40.0		
30–35	54	18.9	4	26.7		
>35	5	1.8	2	13.3		
Total	285	100.0	15	100.0		
**Type of family**	**No**	**(%)**	**No**	**(%)**		
Nuclear	88	30.8	8	53.4	0.06	3.30, df-1
Joint	197	69.2	7	46.6		
Total	285	100.0	15	100.0		
**Religion**	**No**	**(%)**	**No**	**(%)**	0.001	15.07, df-1
Hindu	209	73.3	4	26.6		
Muslim	76	26.7	11	73.4		
Total	285	100.0	15	100.0		
**Caste**	No	(%)	No	(%)	0.74	0.10, df-1
General	126	44.2	6	40.0		
OBC	159	55.8	9	60.0		
Total	285	100.0	15	100.0		
**Educational status**	**No**	**(%)**	**No**	**(%)**		
Illiterate	126	44.2	8	53.3	0.192	7.39, df-5
Primary	36	12.6	2	13.3		
Junior high	49	17.2	2	13.3		
High school	62	21.7	1	6.7		
Intermediate	10	3.5	1	6.7		
Graduate and above	2	0.8	1	6.7		
Total	285	100.0	15	100.0		
**Occupation**	**No**	**(%)**	**No**	**(%)**	0.001	31.81, df-3
Unskilled	15	5.3	6	40.0		
Semi-skilled	213	74.7	4	26.7		
Skilled	45	15.7	3	20.0		
Shopkeeper/Clerical	12	4.3	2	13.3		
Total	285	100.0	15	100.0		
**Socioeconomic** **Status**	**No**	**(%)**	**No**	**(%)**		
Upper class	4	1.4	1	6.7	0.001	20.84, df-4
Upper middle class	4	1.4	2	13.3		
Middle class	22	7.8	4	26.7		
Lower middle class	188	65.9	5	33.3		
Lower class	67	23.5	3	20.0		
Total	285	100.0	15	100.0		
**Parity**	**No**	**(%)**	**No**	**(%)**		
1	86	30.2	3	19.9	0.04	7.92, df-3
2	125	43.8	4	26.7		
3	52	18.3	4	26.7		
≥4	22	7.7	4	26.7		
**Total**	285	100.0	15	100.0		

**Table 4 children-10-00408-t004:** Socio-demographic factors influencing Initiation of breastfeeding.

Socio-Demographic Variables	Initiation of Breastfeeding					
	Early Breastfeeding n = 149		Delayed Breastfeeding n = 151		*p*-Value	Chi-Square Value, Degree of Freedom
**Age of mother** **(in years)**	**No**	**(%)**	**No**	**(%)**	0.001	49.60, df-3
18–23	19	12.7	4	2.6		
24–29	78	52.3	134	88.7		
30–35	48	32.3	10	6.7		
>35	4	2.7	3	2.0		
Total	149	100.0	151	100.0		
**Type of family**	**No**	**(%)**	**No**	**(%)**		
Nuclear	37	24.8	59	39.0	0.008	6.98, df-1
Joint	112	75.2	92	61.0		
Total	149	100.0	151	100.0		
**Religion**	**No**	**(%)**	**No**	**(%)**		
Hindu	125	83.9	88	58.3	0.001	23.89, df-1
Muslim	24	16.1	63	41.7		
Total	149	100.0	151	100.0		
**Caste**	**No**	**(%)**	**No**	**(%)**		
General	122	81.8	10	6.6	0.001	172.38, df-1
OBC	27	18.2	141	93.4		
Total	149	100.0	151	100.0		
**Educational status**	**No**	**(%)**	**No**	**(%)**		
Illiterate	72	35.4	62	47.2	0.001	38.44, df-5
Primary	31	26.1	7	8.9		
Middle	18	10.7	33	18.7		
High school	17	10.7	46	23.8		
Intermediate	9	13.8	2	0.8		
Graduate and above	2	3.0	1	0.4		
Total	149	100.0	151	100.0		
**Occupation**	**No**	**(%)**	**No**	**(%)**	0.03	8.67, df-3
Unskilled	9	6.0	12	8.0		
Housewife	102	68.5	115	76.1		
Skilled	26	17.5	22	14.5		
Shopkeeper/Clerical	12	18.0	2	1.4		
Total	149	100.0	151	100.0		
**Socioeconomic status**	**No**	**(%)**	**No**	**(%)**	0.08	8.22, df-4
Upper class	4	2.6	1	0.6		
Upper middle class	4	2.6	2	1.3		
Middle class	18	12.2	8	5.3		
Lower middle class	87	58.5	106	70.3		
Lower class	36	24.1	34	22.5		
Total	149	100.0	151	100.0		
**Parity**	**No**	**(%)**	**No**	**(%)**	0.007	11.97, df- 3
1	31	20.8	58	38.4		
2	69	46.3	60	39.8		
3	33	22.2	23	15.2		
≥4	16	10.7	10	6.6		

**Table 5 children-10-00408-t005:** Socio-demographic factors influencing new-born bathing practices.

Socio-Demographic Variables	Bathing Practices					
	Early Bathing (n = 123)		Delayed Bathing (n = 177)		*p*-Value	Chi-Square Value, Degree of Freedom
**Age of mother** **(in years)**	**No**	**(%)**	**No**	**(%)**	0.11	5.91, df- 3
18–23	5	4.0	18	10.4		
24–29	87	70.7	125	70.1		
30–35	29	23.6	29	16.8		
>35	2	1.7	5	2.7		
Total	123	100.0	177	100		
**Type of family**	**No**	**(%)**	**No**	**(%)**		
Nuclear	31	25.2	65	36.4	0.03	4.426, df- 1
Joint	92	74.8	112	62.6		
Total	123	100.0	177	100		
**Religion**	**No**	**(%)**	**No**	**(%)**		
Hindu	51	41.4	162	91.5	0.00	88.332, df- 1
Muslim	72	58.6	15	8.5		
Total	123	100.0	177	100		
**Caste**	**No**	**(%)**	**No**	**(%)**		
General	35	28.5	97	54.8	0.001	20.44, df-1
OBC	88	71.5	80	45.2		
Total	123	100.0	177	100.0		
**Educational status**	**No**	**(%)**	**No**	**(%)**	0.00	69.51, df-5
Illiterate	87	70.7	47	26.5		
Primary	12	9.7	26	14.7		
Junior high	9	7.4	42	23.8		
High school	7	5.8	56	31.6		
Intermediate	6	4.8	5	2.9		
Graduate and above	2	1.6	1	0.5		
Total	123	100.0	177	100.0		
**Occupation**	**No**	**(%)**	**No**	**(%)**	0.001	15.81, df-3
Unskilled	10	8.1	11	6.2		
Semi-skilled	88	71.6	129	72.9		
Skilled	13	10.6	35	19.8		
Shopkeeper/ Clerical	12	9.7	2	1.1		
Total	123	100.0	177	100.0		
**Socioeconomic status**	**No**	**(%)**	**No**	**(%)**		
Upper class	1	0.8	4	2.2	0.30	4.83, df-4
Upper middle	2	1.7	4	2.2		
Middle	10	8.2	16	9.0		
Lower middle	74	60.1	119	67.3		
Lower class	36	29.2	34	19.3		
Total	123	100.0	177	100.0		
**Parity**	**No**	**(%)**	**No**	**(%)**	0.01	10.64, df- 3
1	25	20.3	64	36.1		
2	56	45.5	73	41.2		
3	27	22.0	29	16.5		
≥4	15	12.2	11	6.2		
Total	123	100.0	177	100.0		

## Data Availability

Data is Available with the first and second Authors.

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
