# Peer review of "Newborn Care Practices and Associated Factors Influencing Their Health in a Northern Rural India"

_children, 2023, doi:10.3390/children10020408_

Round 1
Reviewer 1 Report
Hi, the submitted article has been reviewed.
The writing of the article is not academic. For example, Study Area is not suitable and the Study setting is not used.
The statement of the problem is very short and does not state the necessity of conducting the study.
The discussion is very limited.
The application of the findings in the clinic has not been stated.
Author Response
Respected Sir,
1. Extensive editing of the English language and style required
The qualified English Professor thoroughly reviewed the English language.
2. Does the introduction provide sufficient background and include all relevant references?
The Introduction was reviewed and cited all relevant references
3. Are all the cited references relevant to the research?
It has been modified and the irrelevant references were modified
4. Is the research design appropriate?
It has been modified
5. Are the methods adequately described
It has been modified and clearly mentioned the methods
6. Are the results clearly presented
It has been modified
7. Are the conclusions supported by the results
It has been aligned with the study results
8. The writing of the article is not academic. For example, Study Area is not suitable and the Study setting is not used.
Ans. It has been revised as per the academic pattern
9. The statement of the problem is very short and does not state the necessity of conducting the study.
It has been modified and mentioned the sufficiently
10. The discussion is very limited.
The discussion has been elaborated on and addressed the objectives
11. The application of the findings in the clinic has not been stated.
It has been mentioned in the Methodology section under the study settings.

Reviewer 2 Report
Thank you for the opportunity to review the work. The authors raise an important aspect of home care for a newborn in environments with limited access to medical care. Unfortunately, the research methodology, content, presentation of results and other aspects of the work require significant changes. Here are the most important issues:
1) Excel sheet is not a dedicated statistical analysis tool, which in my opinion reduces the quality of work
2) Table 1 is illegible, data is not centered. Cross lines make reading difficult. I suggest thematic sorting of the presented data.
3) Figure 1 has no axis write-offs
4) the discussion only presents the results of other studies, and the authors rarely refer to them with their results
5) no study limitations
6) the literature is poor and old
Author Response
Respected Sir,
1. Extensive editing of the English language and style required
The qualified English Professor thoroughly reviewed the English language.
2. Does the introduction provide sufficient background and include all relevant references?
The Introduction was reviewed and cited all relevant references
3. Are all the cited references relevant to the research?
It has been modified and the irrelevant references were modified
4. Is the research design appropriate?
It has been modified
5. Are the methods adequately described
It has been modified and clearly mentioned the methods
6. Are the results clearly presented
It has been modified
7. Are the conclusions supported by the results
It has been aligned with the study results
8. Excel sheet is not a dedicated statistical analysis tool, which in my opinion reduces the quality of work
Thank you for your valuable suggestion. It has been rectified in the statistical analysis section.
9. Table 1 is illegible; data is not centered. Cross lines make reading difficult. I suggest thematic sorting of the presented data.
It has been rectified the typographical errors
10. Figure 1 has no axis write-offs
Fig 1 is a horizontal Bar gram in x-axis numbers and Y-axis the categories was represented
4) the discussion only presents the results of other studies, and the authors rarely refer to them with their results
It has been modified
5) no study limitations
It has been included
6) the literature is poor and old
It has been updated and the relevant latest references were added.

Round 2
Reviewer 1 Report
in material and method section" Study Period and Study Population" delate and And it should be placed under the Study Design heading .
Author Response
Respected Reviewer,
1. Is the research design appropriate?
It has been modified as per your suggestion
2. Are the methods adequately described?
It has been modified as per your suggestion
3. material and method section" Study Period and Study Population" delate, and it should be placed under the Study Design heading.
It has been merged and kept under the subheading of study design
Reviewer 2 Report
I accept the current version. I suggest that the results in the statistically significant table are marked in bold
Author Response
Respected Reviewer,
Thank you for accepting the article.